# Message in a Scaffold: Natural Biomaterials for Three-Dimensional (3D) Bioprinting of Human Brain Organoids

**DOI:** 10.3390/biom13010025

**Published:** 2022-12-22

**Authors:** Pierre Layrolle, Pierre Payoux, Stéphane Chavanas

**Affiliations:** INSERM, Université Paul Sabatier, Toulouse NeuroImaging Center (ToNIC), UMR 1214, CHU Purpan, 31024 Toulouse, France

**Keywords:** biomaterials, brain organoid, 3D bioprinting, scaffold, neural stem cell, induced pluripotent stem cells, disease model, neuron

## Abstract

Brain organoids are invaluable tools for pathophysiological studies or drug screening, but there are still challenges to overcome in making them more reproducible and relevant. Recent advances in three-dimensional (3D) bioprinting of human neural organoids is an emerging approach that may overcome the limitations of self-organized organoids. It requires the development of optimal hydrogels, and a wealth of research has improved our knowledge about biomaterials both in terms of their intrinsic properties and their relevance on 3D culture of brain cells and tissue. Although biomaterials are rarely biologically neutral, few articles have reviewed their roles on neural cells. We here review the current knowledge on unmodified biomaterials amenable to support 3D bioprinting of neural organoids with a particular interest in their impact on cell homeostasis. Alginate is a particularly suitable bioink base for cell encapsulation. Gelatine is a valuable helper agent for 3D bioprinting due to its viscosity. Collagen, fibrin, hyaluronic acid and laminin provide biological support to adhesion, motility, differentiation or synaptogenesis and optimize the 3D culture of neural cells. Optimization of specialized hydrogels to direct differentiation of stem cells together with an increased resolution in phenotype analysis will further extend the spectrum of possible bioprinted brain disease models.

## 1. Introduction

Neurological disorders are the leading cause of disability and the second cause of deaths worldwide, and this burden is expected to be driven up by population aging [1]. Despite considerable progress in medical imaging, the complexity and inaccessibility of the brain still hinder research on the live organ. Post-mortem explorations of brain samples have provided significant insight, but their potential to pathophysiology or drug screening studies is obviously limited. Even though they have provided so much knowledge about brain biology, mouse models do not always properly recapitulate human neurological disorders because of the significant differences in development, structure and physiology of rodent and human brains [2,3]. Classical monolayer (two-dimensional, 2D) cultures of neural cells have unveiled important knowledge on brain disorders with genetic or infectious aetiology (as in the case of lissencephaly [4,5]), but they cannot recapitulate the complex events underlying brain development or homeostasis. Hence, there is an urgent need for new in vitro models. Three-dimensional (3D) cell culture has proven its multiple benefits compared to 2D culture in terms of cell function and homeostasis, and has paved the way for human brain organoids [6]. Since the pioneering work of Lancaster et al. [7], and throughout the last decade, the generation of brain organoids from human induced pluripotent stem cells (iPSCs) or human embryonic stem (ES) cells was a milestone towards modelling healthy or diseased human brain and provided a wealth of knowledge on brain pathophysiology [8]. However, there are still challenges to overcome in making organoids more reproducible and relevant to the complexity of the brain; in particular, brain organoids fail to reproduce cerebral substructures and lack microvasculature [9]. Meanwhile, tremendous advances in 3D bioprinting of live tissues or organs have opened up new horizons for disease modelling in recent years. Three-dimensional bioprinting consists in the precise and automated deposition of cell-laden hydrogels, so-called “bioinks”, for the biomanufacturing of complex human living tissues or organs, including neural tissues and, expectedly, brain. Convergence between organoid technology and 3D bioprinting is expected to open new avenues in brain research. The development of optimal hydrogel biomaterials for the bioprinting of neural organoids is of utmost importance and has been the subject of an increasing amount of work in recent years. A whole field of research has improved our knowledge about biomaterials both in terms of their intrinsic properties and the impact of their use on 3D brain cells and tissue culture. A wealth of recent and very informative articles have cleverly reviewed brain mechanobiology [10] or mechanical properties of biopolymers [11,12,13]. Meanwhile, there is a lack of review articles that focus on the biological roles that these biomaterials exert on the enladen cells. Biomaterials are rarely biologically neutral. they are able to deliver or collect biological signals to or from the cells, provide cells with adhesion sites and shape cellular microenvironments. Hydrogel biomaterials direct cell differentiation depending on their matrix stiffness and may potentially induce organogenesis through mechanotransduction [14,15]. Furthermore, the mechanical properties of the cell microenvironment are involved in normal brain tissue function but also in neuropathological situations [10]. Although of crucial importance, matrix elasticity has been poorly investigated in 3D culture of brain organoids, as these were mainly produced using Matrigel^TM^, a basement membrane matrix secreted by mouse sarcoma cells. It is hypothesized that defined hydrogel biomaterials may provide a more reproducible cellular microenvironment to direct stem cell proliferation and differentiation than animal-derived and variable extracellular matrices to produce brain organoids. In combination with 3D bioprinting, the multiscale complexity of brain structures may be mimicked. We here review the current knowledge on biomaterial scaffolds amenable to support 3D bioprinting of neural tissues or organoids with a particular interest on the biological dimension of their role.

## 2. Brain Organoids: Potential and Limits

Brain organoids are 3D clusters of cell populations derived from primary tissue, embryonic stem cells (ESCs), or induced pluripotent stem cells (iPSCs), capable of self-renewal and self-organization, and that recapitulate certain organ functionality [16]. Brain organoids harbour brain-like substructures, allow for neuronogenic or gliogenic differentiation, and exhibit electrophysiological activity indicative of neuronal network level functioning [17]. Organoids generated from induced pluripotent stem cells can model complex neurological disorders such as those from patients with Rett syndrome, which show abnormal, epileptiform-like activity [18]. This makes organoids invaluable tools for pathophysiological studies or drug screening. Since the first human brain organoid described by Lancaster [7], several studies have pushed the limits of what was thought possible. Neurons represent a non-homogeneous network of cell populations with molecular, regional, and functional specificities showing different sensitivities to disease. Specific procedures, so-called “guided” procedures, have been reported to drive organoid patterning towards distinctive lineages [19]. For instance, organoids with high density in hippocampal [20], cortical [21], dopaminergic [22], glutaminergic [23], gamma-aminobutyric acid (GABA)-ergic [23] neurons, as well as retinal ganglionic neurons [24], astrocytes [25], microglia [26,27] or oligodendrocytes [28], have been produced, to mention a few ground-breaking examples. Furthermore, organoids which recapitulated bilateral optic vesicles [29] and vasculature-like structures [30,31] have been reported.

Against all odds, the fascinating ability of organoids to organize themselves spontaneously raises true issues [9]. Firstly, scalability issues hamper their use in high-throughput assays such as drug screening, since the volume of organoids can exceed one cubic millimetre upon maturation [32]. Moreover, organoid spatial organisation is largely unpredictable and thus hampers the reproducibility of the 3D models. Furthermore, while the diversity and distribution of cell types in organoids have strong similarities to those of embryonic or foetal tissue, the spatial organisation of cellular components and paracrine signals remain far from nature. In particular, axial patterning of soluble morphogens requires topological patterns that simply do not exist in organoids. Organoids cultured in suspension exert no mechanical constraint driving neural tissue development, as is the case in vivo (for example as imposed by the developing skull) [33]. Inconsistency might result in heterogeneity and phenotypical variability of organoids, possibly overlapping or even hiding caused by the disease modelled [34]. Controlled assembly of pre-differentiated organoids [35] is a clever approach to bypass this problem, but it offers limited room to achieve standardization of a model amenable to reproducible assays or controlled geometries. Furthermore, the innermost cell populations within organoids hardly have access to oxygen and nutrients present in the culture medium, which inevitably results in local necrosis with possible release of soluble mediators impacting the rest of the organoid. This is strongly contributed by the absence of vascularizing structures, which negatively impacts progenitor populations [36] and further contributes to experimental variability [34]. Current strategies to overcome this pitfall rely on culture in bioreactors, transplantation into mouse models or perfusion with microfluidic devices [37].

## 3. Three-Dimensional Bioprinting: Benefits and Challenges

Three-dimensional (3D) bioprinting is a highly promising technology for both tissue regeneration and organ replacement but may certainly help in the short term to develop models for studying human organ development and diseases as well as for drug screening in vitro as alternatives to animal experiments [38]. Briefly, bioprinting consists in the computerized fabrication of 3D structures composed of living cell-laden biocompatible hydrogels layer by layer. Three-dimensional bioprinting allows different cell types to be distributed and arranged in any possible pattern, offers high reproducibility since it is computer-driven, and favours accessibility of cells to nutrients and oxygen because it controls the pattern infill in the 3D tissue engineered constructs. It also allows for pre-differentiation of cell populations before controlled assembly by using bioprinting, a so-called “bottom-up” approach. For instance, 3D bioprinting allows for the accurate picking and assembly of whole cell spheroids into higher-order structures with minimal cell damage [39]. Furthermore, 3D bioprinting offers a simple solution to reproduce gradients of morphogenic molecules such as those involved in development, by bioprinting cell populations in distinct but contiguous hydrogels containing the appropriate soluble factors at different concentrations. Finally, 3D bioprinting makes it possible to insert structures recapitulating blood vessels or even vascular networks inside an organoid, by using sacrificial bio-inks, which are intended to dissolve to leave room for the “vessel” lumen [40].

Thus far, 3D bioprinting of brain organoids has been limited by technological challenges such as printability of hydrogel bioinks, shape fidelity of 3D constructs post fabrication and limited cell migration and/or differentiation [41,42,43]. Its development is the subject of much research at the technological level but also in terms of the hydrogels used as bioinks. These biomaterials should not only be cytocompatible to support cell adhesion, growth and differentiation, but they must have adequate viscoelastic properties for a continuous micro-extrusion as well as forming cross-links to ensure the stability of the 3D constructs in culture [44]. Most hydrogels having favourable biological properties to support brain organoids culture do not meet the physicochemical requirements for 3D bioprinting (e.g., native collagen, fibrinogen, Matrigel). For instance, extrusion bioprinting requires shear shinning and high viscosity properties to ensure a continuous and stable strand extrusion of the bioink at physiological temperature, pH and osmolarity. For these reasons, blends of hydrogels are often used to improve shear thinning properties of bioinks. They result, however, in lower ability to support neurogenesis and brain organoid maturation. Furthermore, components of the brain extracellular matrix are often chemically modified by methacrylate coupling (e.g., gelatine Methacrylate, GelMA) to form cross-links by photopolymerization and to render the 3D constructs stable in culture. Nevertheless, photocurable bioinks induce some cell mortality due to the presence of free radicals. A large number of solutions have been tested in recent years to improve bioink properties in terms of printability, long-term stability, cell compatibility and organoid maturation, using raw or defined, homogeneous or heterogeneous biomaterials, should they be chemically modified or not. Recent works have revealed that unmodified polymers can efficiently support bioprinting of neural tissues, and we here focus on such native molecules. Most of them are components of the brain extracellular matrix, a key structure to understand the outcomes of the microenvironment on cell and tissue homeostasis.

## 4. The Brain Extracellular Matrix

The extracellular matrix (ECM) is a dynamic three-dimensional network of macromolecules that provides structural support for cells and tissues [45]. Decades of research have documented the paramount role of ECM in cell fate and behaviour, in developing [46] or adult [47] tissues. The critical role of brain ECM is supported by the fact that it is altered in neurodevelopmental disorders [48] and in Alzheimer’s, Parkinson’s, and Huntington’s diseases [49]. ECM is prominent in the human foetal brain, accounting for 40% of brain volume, and particularly abundant around the earliest born cortical neurons, at the crossroads of axonal pathways, at the developing cortical–white matter interface, and in the marginal zone [48]. Adult brain ECM has a unique composition in the human body, with the prominence of proteoglycans, tenascins and hyaluronic acid that highly bind water in the almost absence of fibrous proteins abundant in other tissues (collagen, fibronectin, vitronectin) [6,50]. To note, PeriNeural Networks (PNN) are specific microenvironments which wrap neuron cell bodies, dendrites and the initial segments of axons [45]. ECM has a critical role in the orchestrated events which shape the human brain, as cell proliferation, migration and differentiation, or axonal outgrowth, axonal and dendritic growth cone dynamics and synaptogenesis ([48] and references therein). The brain ECM also plays a key role in neural stem cell maintenance, proliferation, and differentiation [51]. Importantly, specific to the brain parenchyma are synaptic ECMs which tightly wrap and dynamically stabilize synapses in relation with their plasticity [52]. The brain is also an extremely soft tissue, as neural ECM has low elastic modulus (approx. 110 Pa and 1 kPa for neonatal and adult brain, respectively) and large porosity, as compared to other tissues [13]. Consistently, ECM viscoelasticity has been shown to significantly impact cell behaviour ([53] and references therein). Young’s moduli higher than 1 kPa or lower than 500 Pa were found to favour gliogenesis or neuronogenesis, respectively [13]. More recently, it has been shown that low elastic moduli (approx. 300 Pa) favoured the differentiation of human iPSCs into forebrain-like neurons, whereas higher elastic moduli (approx. 1 kPa) promoted differentiation into hindbrain-like neurons [54] Forebrain, midbrain and hindbrain are three specific regions arising from the progressive segmentation of the brain along the rostral–caudal axis during development [55]. Yet, the effects of biomaterials on the regional identity of the produced neurons is an important issue to address if relevant disease models are to be generated. For instance, forebrain neurons such as cortical excitatory glutamatergic neurons, inhibitory striatal medium spiny neurons or GABA interneurons are involved in autism spectrum disorders, schizophrenia [56] Alzheimer’s [57] or Huntington’s [58] diseases. The midbrain contains dopaminergic neurons involved in Parkinson’s disease [59]. The hindbrain contains sensory-motor neurons involved in breathing, vision, mastication, equilibrium and locomotion [60], and serotoninergic neurons involved in mood, circadian rhythm, appetite, cognition and addiction [61]. The so-called rostralizing or caudalizing factors are those which favour forebrain- or hindbrain-like neurons within the organoid, respectively. Thus, that study suggested that stiffer hydrogels have a caudalizing effect on neuron fate [54]. Altogether, these works support the invaluable potential of ECM components for 3D neural cell or tissue culture. They also suggest that biomaterials resembling foetal brain ECM rather versus adult brain ECM would best support neural organoid genesis, consistent with the fact that organoid genesis recapitulates development. Yet, foetal brain ECM is subjected to extensive qualitative and quantitative changes throughout development [6,62], which makes it difficult to predict which biomaterial would best support the generation of cerebral organoids. Nevertheless, a significant body of work has provided highly helpful insight on the impact of ECM components on neural tissue culture. 

## 5. Scaffold Hydrogel Biomaterials

Here, we refer to natural scaffold polymers to denote biomaterials capable of forming coherent, bioprintable hydrogels on their own, by simple physical or chemical reaction, without undergoing covalent modifications. Those used for 3D culture of neural cells or organoids are either extracellular matrix extracts or single defined biopolymers.

### 5.1. Scaffolds from Extracellular Matrix Extracts

Matrigel^TM^ is a blend of ECM proteins derived from Engelbreth–Holm–Swarm mouse sarcoma cells [63], containing more than 1800 unique proteins [64] including laminin (55%), collagen IV (30%) and entactin (6%) [6]. Matrigel^TM^ is one of the most common biomaterials used as a 3D hydrogel scaffold. Unlike gelatine, Matrigel shows a gelling temperature compatible with physiological or culture conditions. Matrigel is renowned to provide a favourable environment to neural cell culture, and most brain organoids have been generated from stem cell clusters encapsulated within a Matrigel^TM^ drop [7,65]. A variety of neurological conditions modelled using induced pluripotent cells from patients has been reported using Matrigel^TM^, which further underscores its merits ([6] and references therein). Yet Matrigel^TM^ has a number of drawbacks. Significant batch-to-batch variability in Matrigel^TM^ composition has been evidenced [64]. Its undefined composition makes it difficult to identify the signals actually influencing organoid structure and function. Matrigel^TM^ may contain sub-optimal amounts of some ECM components, as shown for laminin-511 [66], or variable blends of growth factors possibly biasing interpretation of experiments [67]. The mechanical properties are heterogeneous within a Matrigel^TM^-based object [66]. Matrigel^TM^ crosslinks at room temperature and thus requires a temperature-controlled extrusion printing system. Furthermore, its linear elastic behaviour results in complex handling and poorly controlled mechanical properties [68,69]. It is not possible to extrude Matrigel^TM^ by using pneumatic extrusion, as it is ejected out of the syringe with an uncontrollable behaviour as soon as pressure is applied [70]. Matrigel^TM^ has been cross-linked with collagen or polyethylene glycol to overcome these drawbacks, but these mixed hydrogels display a dense, poorly organized architecture that yield highly heterogeneous scaffolds [68]. Together, these findings highlight the limitations of Matrigel^TM^ for 3D bioprinting. Only recently were successful bioprinted pure Matrigel^TM^-based free-standing structures with good shape fidelity, by using a deeply customized printer originally dedicated to fused deposition modelling of thermoplastic polymers, reported [70]. Lastly, the murine tumour origin of Matrigel^TM^ raises the question of its immunogenicity and biosafety, which hampers its potential in human clinical applications.

Another ECM-based hydrogel base is decellularized brain ECM (dECM). dECM extracted from animal tissue is meant to provide cell cultures with a molecular microenvironment as close as possible to that of the tissue of origin. dECM extracts from adult or foetal pig brain were shown to contain glycosaminoglycans, collagen I, III–VI, perlecan, laminin and the growth factors basic fibroblast growth factor (bFGF) and nerve growth factor (NGF), and to support neural cell growth, neurite outgrowth and neuronal network formation when used as a coating agent in 2D cultures [71,72,73,74]. dECM from different sources has been used for bioprinting a variety of tissues, but never for brain-related ones, making its relevance for brain organoid bioprinting uncertain at this time. dECM presumably also has great potential relative to the heterogeneity of the brain parenchyma. ECM composition is variable upon brain sub-structures [75]. Therefore, assessing whether ECM extracted from specific brain substructures promotes the differentiation of progenitor cells into specific neuronal subtypes (such as, for example, ECM from substantia nigra and dopaminergic neurons) could be an advance for directed differentiation of bioprinted progenitors and for bottom-up approaches. dECM, however, has the same drawbacks as Matrigel^TM^ in terms of printability, reproducibility and biosafety.

### 5.2. Scaffolds from Single Polymers: Agarose and Alginate

Single polymers overcome some problems posed by matrix extracts, resulting in well-defined hydrogels that are not derived from animal tissues and can be obtained in large amounts and at high purity levels. The natural polymers that have been used for hydrogels supporting the 3D culture of neural cells, tissues or organoids are agarose and alginate. Agarose, a natural polysaccharide derived from red seaweed, was the first to be investigated for the culture of neural cells, as early as 2001 [76]. Agarose does not allow for proper cell attachment [77] and has poor biodegradability in mammalian tissues, which limits its potential use in biotherapy [78]. To the best of our knowledge, one single article [79] reported the feasibility of producing functional neural tissue containing glial cells and neurons that were predominantly GABAergic, using an agarose-based bioink, but in combination with a much more widely used polymer as compared to agarose: alginate [80]. Alginate-based hydrogels have provided numerous and impressive results. Alginate is a linear polysaccharide from algae composed of (1-4)-linked β-d-mannuronic acid and α-l-guluronic acid (M and G, respectively). Alginate can be ionically crosslinked into a gel form through exposure to divalent cations, such as Ca^2+^, at physiological pH and osmolarity [81]. Notably, this is a reversible process that allows for gel dissociation and cell recovery upon exposure to chelators as EDTA or citrate, a valuable feature in the context of a bottom-up approach. Other merits of alginate hydrogels are the absence of cytotoxicity, mechanical properties close to those of mammalian tissue extracellular matrix and insensitivity to biodegradation by the embedded cells since mammal cells lack the alginate-degrading enzyme, namely alginase [81,82]. The viscosity of alginate hydrogels depends on the average molecular weight and molecular weight distribution, the average G to M ratio and the concentration of alginate, along with the pH of the solution [83]. It was found to be highly variable, spanning almost four orders of magnitude, and to adversely affect the survival of U87-MG-transformed cells when greater than 120 Pa.s [80]. It was reported that bioprinting of self-supporting structures with good structural integrity required 4% (*w*/*v*) alginate hydrogel with final CaCl_2_ concentration of 40 mM [80], a titre much greater than that allowed for neural cell culture (<2 mM).

A bundle of studies demonstrated that 1% alginate shows structural similarities to hyaluronic acid, has an elastic modulus comparable to brain tissue, and efficiently supports neural lineage growth or differentiation ([84] and references therein). Notably, a dramatic milestone was achieved in 2021, which reported the efficient generation of human iPSC-derived functional dopaminergic neurons embedded in alginate beads, with long-term (50 days) maintenance of phenotype and function, and without detectable necrotic nuclei [59]. This spectacular achievement confirmed the role of alginate as an excellent scaffold for hydrogels dedicated to neural cell culture. However, the alginate base was supplemented with fibronectin in that study. Alginate-based hydrogels show two drawbacks: they poorly support cell adhesion and have limited long-term stability in culture medium, because monovalent cations progressively replace divalent cations in the polymer [81]. These two roadblocks make it necessary for alginate hydrogels to be supplemented with adjuvants to increase stability, provide cells with adhesion and anchorage sites and, finally, to recapitulate the native cellular microenvironment as closely as possible.

## 6. Natural Adjuvants

Here, we mean adjuvants to denote single or blended polymers or macromolecules that do not have the intrinsic mechanical ability to serve as a scaffold in hydrogel but that provide physical or biological assets to support and optimize 3D cell culture. In addition to their mechanical properties, natural adjuvants may feature motifs that play a role in growth factor availability or in cell functions, such as adhesion, motility, and synaptogenesis, mediated by outside-in and/or inside-out cell signalling (Figure 1).

### 6.1. Collagen

Collagen refers to a superfamily of almost 30 glycoproteins characterized by the amino acid repeat [Gly–X–Y]*_n_*, where X and Y are proline and hydroxyproline, respectively, and which are folded as a triple helix. Collagen polypeptides may assemble into fibrils, beaded filaments, anchoring fibrils, and networks depending on their composition [85]. Collagens are ligands of integrins. Integrins are critical cell-adhesion and signalling transmembrane receptors which exert pleiotropic roles in differentiation and maintenance of neural stem cells, axon outgrowth, dendrite branched morphogenesis, and synapse regulation in the developing brain [86]. Integrins are heterodimers composed of α and β subunits. Expression of integrins α1, 3-8 and β4, 5 is regionally differentiated in the adult brain and is detailed elsewhere [87]. Expression of subunits α3/5/6/7/v, and β1 and β4 has been detected in neural stem or progenitor cells [88]. Integrins α5β1 and αvβ1, to which collagen binds, have been shown to have a key role in brain development [85,86]. Collagens I–III are also ligands of the two important regulators of myelination and oligodendrocyte homeostasis, namely the discoidin receptors DDR1 and DDR2, which connect to the cytoskeleton [89].

Collagen hydrogels have linear elasticity, and their physicochemical properties, from compression modulus or viscosity to fibre structure and dynamics, depend widely on the source of tissue, the solubilization method and the pH, ionic strength and concentration of the collagen solution as it has been deeply documented elsewhere [90,91]. Collagen maintains a liquid state below 37 °C and is rarely used as a base bioink on its own due to its mechanical instability and slow gelation rate at physiological temperatures, which limit its ability to hold its shape once extruded [41].

Collagen emerged early on as a promising adjuvant. Almost 20 years ago, neural progenitor cells from embryonic rat brains were embedded within type I collagen in the presence of bFGF and showed active expansion of progenitors and efficient production of neurons which exhibited cell polarity, expression of neurotransmitters, ion channels and receptors, excitability, and active synaptic vesicle recycling [92]. In a toroidal model of primary rat cortical neuron culture, collagen was reported to be mandatory for specific and efficient axon growth in the central area devoid of cell bodies, whereas the ring-shaped scaffold containing silk fibroin and collagen efficiently supported cell function [93,94,95]. The collagen/silk-based hydrogel showed good cytocompatibility and the relevance of the model was supported by its ability to exhibit molecular and electrophysiological responses to axonal injury similar to those observed in vivo [93]. Collagen/silk-based scaffolds were much more stable in time than fibrin/silk- or Matrigel/silk-based scaffolds (months versus weeks, respectively), whereas hydrogel containing only collagen dramatically shrank in height within days [93]. Subsequent studies showed that supplementation of collagen/silk hydrogels with dECM from adult or especially foetal porcine brains significantly improved cell viability and function and neuronal network formation [73]. Taken together, these works revealed that collagen provided a stable and efficient support for both axon growth and cell function when associated with silk fibroin and brain dECM. Importantly, such hydrogels efficiently supported cell growth, differentiation and function in 3D cultures of human neural tissue from brain surgical resections [96] or iPSCs derived from healthy donors or Alzheimer’s and Parkinson’s disease patients [97], which underscore their potential in human neurological disease modelling. Consistently, a recent study demonstrated that blended alginate/collagen hydrogels promoted neurogenesis and neuronal maturation from human iPSCs [98].

### 6.2. Fibrin

Fibrin is a natural polymer formed from hydrolysis of its zymogen form, fibrinogen, by thrombin during the blood coagulation cascade. Fibrin monomers are covalently crosslinked by Factor XIIIa to assemble into a fibrous network with physical properties resembling soft tissue, and which can be hydrolysed by plasmin. Fibrin can be obtained from the plasma of a variety of species (salmon, rat, bovine, human) or from human platelet lysate preparations (PLP) after exposure to thrombin.

Fibrin is a viscoelastic polymer with nonlinear elasticity and high viscosity, and the latter makes extrusion of fibrin hydrogel challenging [99]. The physicochemical properties of fibrin have been deeply studied elsewhere [100]. Fibrin filaments form soft networks that allow for a high degree of deformation without breakage [100]. The mechanical properties of fibrin hydrogels can be modulated by altering the concentration of fibrinogen, Ca^2+^ and thrombin present during polymerization, and optimized values have been proposed for neural stem cells [101]. Fibrin is definitely a bioactive molecule as much as a structurant polymer. As collagen, fibrin can bind integrins through the peptide motif Arg-Gly-Asp (RGD). In addition, fibrinogen and fibrin contain heparin-binding domains which stably bind a variety of growth factors from the FGF, Platelet Derived Growth Factor (PDGF)/Vascular Endothelial Growth Factor (VEGF), Transforming Growth Factor (TGF) -β and neurotrophin families for which they are key transporters and relayors to cells [102]. Growth factors have been found in non-concentrated human PLP in the scale of 10 µg/mL (FGF), 1 µg/mL (PDGF), 0.1 µg/mL (Insulin-like Growth Factor (IGF-1)) and 0.1 ng/mL (Epidermal Growth Factor (EGF), TGF-β, VEGF) [103] and may be carried into hydrogels containing fibrin from PLP. Further investigations are needed to decipher the outcomes of the presence of blended growth factors on neural cell culture. Fibrin hydrogels have been shown to support mouse or human NSCs viability, efficient differentiation into GABA-ergic or dopaminergic neurons and/or neurite outgrowth, with or without electrophysiological abilities [104,105,106], although one study reported that fibrinogen inhibits neurite outgrowth via β3 integrin-mediated phosphorylation of the EGF receptor [107]. Salmon fibrin was shown to exhibit better support to neurite outgrowth than that from mammalian species [108]. Fibrin gels prepared with high fibrinogen concentrations (8–10 mg/mL) were observed to associate with lesser levels of neurite extension and neuronal differentiation, as compared to those prepared with lower fibrinogen concentrations (6 mg/mL), possibly owing to their smaller pore area and/or higher rigidity [104]. This finding was partly consistent with an earlier study which revealed that fibrinogen exerts a concentration-dependent inhibition of neurite outgrowth, although at a much lower concentration range (1.5 mg/mL) [107]. To note, physiological concentration of fibrinogen in non-concentrated human PLP was assessed to be in the 100 µg/mL range [109], a level which seems far enough lower than the previous ones not to impair neurite outgrowth. Recently, a noteworthy study revealed that fibrinogen induces murine neural stem cell differentiation preferentially into astrocytes via BMP signalling [110]. This makes fibrin/fibrinogen particularly interesting given that astrocyte differentiation from neural progenitors is known to be a more tedious and less efficient process than neuronogenesis [111]. Lastly, it has been shown that combined collagen and fibrin provided a suitable substrate for a three-dimensional culture matrix for neuronal survival and differentiation [77]. Fibrin rapidly degrades in vitro and hardly lasts longer than two weeks in culture, although the presence of aprotinin, a serine protease inhibitor, or of genipin, a small molecule from gardenia fruit, slows its degradation to some extent [101,105].

### 6.3. Gelatine

Gelatine is the product of thermal denaturation of collagen and has been widely used in 3D bioprinting [82,99]. The viscosity of gelatine solutions is highly sensitive to the temperature and to gelatine concentration. They undergo gelation at low temperatures (<30 °C). High viscosity gelatine solutions are not compatible with extrusion bioprinting unless modulation of temperature lowers their viscosity. This low gelling temperature may adversely affect the viability of cells throughout the process of bioprinting. More importantly, it makes gelatine progressively vanish in culture, as it dissolves in medium at 37 °C. Finally, because gelatine is of animal origin, it does not fill the biosafety requirements for eventual therapeutical applications. Gelatine significantly improves the process of bioprinting because its viscosity improves the accuracy of deposition and shape fidelity, and because it rapidly imparts physical stability to the bioprinted object. Hence, it remains a very valuable adjuvant helper for the bioprinting step.

### 6.4. Hyaluronic Acid

Hyaluronic acid (HA, aka hyaluronan) refers to linear-chain polysaccharides composed of repeating disaccharides of *N*-acetyl-glucosamine and glucuronic acid, connected exclusively by beta-linkages, with high conformational diversity and variability in molecular weight [112]. HA of different molecular weights can be obtained from bacterial (with possibly a high degree of purity), animal (bovine vitreous humour, rooster comb, shark skin) or human (umbilical cord) origin [113]. 

Unmodified HA has a slow gelation rate; it lacks the mechanical strength and intrinsqual stability to function as an independent bioink [99]. Rheology and physicochemical characteristics of a variety of hyaluronic acid hydrogels used in surgery have been described recently [114].

HA is necessary for such early events in the development of neural crest cell migration [115]. HA is a prominent component of the ECM and is particularly abundant in the foetal brain [116], where it has been proposed to be a central organizing molecule around which the other ECM components fall into place [45]. HA has been shown to be critical in the neural stem cell niche [117] where it regulates neural progenitor/stem cells in a fashion dependent of CD44, a membrane receptor of which HA is a ligand [118]. HA also binds to the Receptor for Hyaluronic-Acid-Mediated Motility (RHAMM), and to the Intercellular Adhesion Molecule 1 (ICAM-1), which are expressed in the brain [54,115]. HA binding to CD44, RHAMM and ICAM-1 has been shown to activate the intracellular signalling pathways ras, src, and erk, respectively, which in turn positively regulate cell growth, motility and adhesion, respectively [119]. An impressive number of studies have reported that HA plays key roles in neuronogenic or gliogenic differentiation, neuron and astrocyte migration, neurite outgrowth, axonal pathfinding and astrocyte activation [84,120,121]. In addition and importantly, a recent study has disclosed that HA regulates synapse formation and the balance between excitatory to inhibitory signalling in developing neural networks [122]. It has been proposed that abundant HA within the ECM results in a low elastic modulus comparable to that of the developing brain, and thus could be beneficial to neural differentiation [84]. Molecular weight has an impact on the biological outcomes of HA. HA with molecular weights ranging from 20 to 200 kDa is rather involved in embryonic development [113], whereas high-molecular-mass (greater than 200 kDa) HA promotes cell quiescence, causes cell cycle arrest, and protects against apoptosis and oxidative stress, which is of importance since neurons are particularly sensitive to reactive oxygen species [112]. HA shows high porosity which allows for easy diffusion of nutrients and gas in culture [99]. On the other hand, as alginate, unmodified HA does not promote cell adhesion by itself [99]. Recent works have shown the benefits of HA for 3D neural tissue culture from either progenitors or iPSCs, in association with collagen and two alternative organic polymers: polyvinyl alcohol [123] or poly(ethylene glycol) diacrylate [124], respectively. HA was also used as a hydrogel to generate an innovative and efficient granular hydrogel where human iPSCs grew and underwent neuronogenic differentiation at the discrete boundaries between granules [125]. Although HA lacks the mechanical properties to function as an independent bioink, these studies together indicate that it is a highly beneficial adjuvant for neural tissue culture.

### 6.5. Laminin

Laminins are cross-shaped heterotrimeric αβγ glycoproteins that constitute the major non-collagenous component of basement membranes (BM) [45]. Laminins are implied in the most early events in development, from peri-implantation onwards, including neurulation [126,127]. Laminins are pivotal in neural cell homeostasis and function. As collagen, laminins are ligand for integrins (in particular integrins α3/5/6/7 β1 and α6 β4) [128]. Laminins also bind the cell-surface receptor α-dystroglycan [129], which is itself connected to the actin cytoskeleton, which exemplifies the pivotal role of laminins in outside-in cell signalling. Furthermore, α-dystroglycan was shown to regulate the trajectories of extending axons throughout the mammalian brain [130], and studies on α-dystroglycanopathies have revealed that it exerts essential roles in brain development, including cortex formation and myelination, as well as in neural cell migration and polarization [131]. In addition to triggering signal transduction, laminins use self-interactions to form supramolecular polymers prone to bind other ECM molecules such as fibronectin, nidogen and perlecan, thus forming a biomechanical and signalling continuum within ECM [128]. Insights from human genetic diseases and mouse models with laminin deficiency have disclosed significant roles for laminins in brain cell functions and differentiation [128]. Laminin has been used as an absolutely essential substrate for 2D culture of neural stem cells or neurons for years, until recently [4,88]. Laminin has been shown to promote proliferation, survival and migration of human NSCs in an integrin-dependent manner, and to enhance neuronogenic differentiation [132]. It has also been reported to be a beneficial adjuvant for a variety of hydrogel scaffolds to support 3D culture of neural cells, favour neuronogenic and gliogenic differentiation and improve neural progenitor transplantation into the injured brain [133]. Consistent with a specific role in neuron function, laminin was shown to specifically enhance neurite outgrowth from human iPSC-derived neurons when associated with fibrin and HA, as compared to the same blend without laminin [134]. Moreover, laminin seems to be beneficial to oligodendrocyte differentiation from human iPSCs or umbilical cord blood cell-derived neural stem-like cells or rat neural stem cells, than other coating materials [133,135]. Structure–function studies have provided insight on the biological role of laminin. Several short peptides present within laminin polypeptide sequences, including RGD, Ile-Lys-Val-Ala-Val (IKVAV, within α chains) and Tyr-Ile-Gly-Ser-Arg (YIGSR, within β chains) were shown to support adhesion and growth of neural progenitors, given that their respective concentrations are optimized [133], whereas the dodecapeptide Ac-Cys-Cys-Arg-Arg-Ile-Lys-Val-Ala-Val-Trp-Leu-Cys (CCRRIKVAVWLC) was shown to support attachment, growth, and long-term culture of human neural stem/progenitor cells when conjugated to a polyethylene–glycol scaffold [136]. These studies have opened the way to engineering efficient, stable and reliable synthetic peptides prone to substitute laminin extracted from biological sources.

### 6.6. Other Natural Adjuvants: Fibronectin, Chitosan and Tenascin

Other biopolymers are potentially beneficial adjuvants for 3D culture of bioprinted neural tissue but have been the subject of less investigations with neural cells than previous ones. Further research is therefore required to assess their actual potential in the context of bioprinted neural organoids. These are fibronectin, chitosan and tenascin.

Fibronectin (FN) is a fibrous, V-shaped, dimeric ECM glycoprotein that binds integrins [45]. Developmental studies in various animals models had shown that FN plays a major role in cell survival, adhesion and migration, as well as neurite outgrowth and synapse formation [137]. Fibronectin showed beneficial outcomes on the growth or differentiation of neural tissue from human iPSCs [59], which were not observed with mouse ES cells however [84], and further investigations are thus needed to decipher its promising, yet unconfirmed, adjuvant role.

Chitosan is a linear polysaccharide derived from deacetylation of the arthropod polymer chitin. It shows biocompatibility, nontoxicity, and biodegradability, and is a strong candidate as a supporting polymer in central nervous system repair strategies, despite its high viscosity and low solubility at neutral pH [138]. Another potential drawback of chitosan is its possible neurotoxicity as a polycationic polymer [139], even though it has been reported as neuroprotective in the context of neurological disorders [140]. Chitosan was observed to be beneficial to NSCs growth and function in a bundle of studies (reviewed in [138]). Chitosan was also shown to enhance neuron adhesion in agarose gel, although it may alter neuron morphology [78]. Chitosan has also been associated with concentrated (5%, *w*/*v*) alginate and agarose to efficiently print neural tissue [79]. The weak mechanical strength of chitosan and its poor solubility at neutral pH limit its use for bioprinting [99]. An improved procedure for the preparation of a chitosan bioink amenable for bioprinting scaffolds at high resolution (30 µm) and with high shape retention was described elsewhere [141].

Tenascin-C is a homohexameric, star-shaped glycoprotein which harbours multiple epidermal growth factor (EGF)-like repeats, fibronectin type III-like repeats and a C-terminal fibrinogen-like domain. It is secreted by neural progenitors (radial glia cells), astrocytes and neurons throughout development [142]. It has a considerable importance in brain development, specifically within the microenvironmental niche of neural stem cells, of which regulate self-renewal, proliferation and differentiation [142]. To the best of our knowledge, only a couple of studies have queried the potential of tenascin-C as an adjuvant for neural tissue engineering, and both concluded its positive and stimulatory effects upon survival and proliferation of mouse NSCs, in combination with either HA [143] or RGD-conjugated polyethylene glycol [144]. One limitation of tenascin-C is its poor disponibility and high cost.

## 7. Simple Adjuvant Molecules

Simple natural or synthetic molecules may also be of interest to improve bioink performance for 3D bioprinting brain organoids. A recent work revealed that genipin exhibits neuritogenic and neuroprotective properties and enhances neuronal differentiation of neural progenitors derived from human iPSCs in the presence of fibrin [105]. The polycationic homopolymer poly-L-ornithine (PO) has been shown to better promote proliferation, neuronogenic differentiation, filopodia formation and migration of rat neural progenitor cells when coated on a 2D surface substrate, as compared to poly-L-lysine and fibronectin [145]. PO could thus be an interesting adjuvant to test for 3D neural tissue culture given that its concentration remains far enough from the threshold of cytotoxicity of its soluble form [146]. The Rho kinase inhibitor Y-27632 has been used for to support viability after thawing of sensitive cells such as ESCs, iPSCs or neural stem cells. A recent study has revealed that continuous Y-27632 exposure efficiently supported the growth of iPSC-derived neural populations in alginate beads, without a detectable deleterious effect, even in long-term culture [59].

## 8. Discussion

An abundant bibliography underlines the potential of biomaterials for the 3D bioprinting of neural tissues or brain organoids. Here, we focused on unmodified biomaterials, but chemical modification of natural polymers is also a vast and well-documented field of research.

Studies converge to point to alginate as a particularly suitable bioink base for neural stem cell encapsulation and maintenance in culture. Its qualities of flexibility, stability and biocompatibility make alginate a preferred scaffold today for cell encapsulation but its long-term stability in low calcium media and the absence of cell binding sites require blending with other hydrogels. Gelatine appears to be an ephemeral but valuable helper agent for efficiency and accuracy of bioprinting given that its sensitivity to temperature is rightly managed. Adjuvant molecules form a secondary network with an actual role on the enshedded cell (Figure 2).

Care must be taken before drawing formal conclusions about the precise role of each macromolecule in this sophisticated network. First, as shown in this review, the biological effect of a biomaterial upon neural cell homeostasis and function may depend on its concentration (e.g., fibrin), its molecular form (e.g., HA molecular weight), and especially its associated partners when blended. Moreover, numerous studies report the effectiveness and relevance of biomaterial cocktails and their applications, without it being possible to discriminate the specific role of a particular component of the blend, such as in the study by Arulmoli et al., who rightly compared the effect of a fibrin/HA mixture with or without laminin [134]. Nevertheless, mixed hydrogels have shown significant potential for the bioprinting of neural tissue. For instance, silk fibroin combined with collagen could effectively support both the bioprinting of stable models and the function of the neural tissue they embed. It is likely that new combinations of unmodified biomaterials will open new avenues for 3D bioprinting in the future.

Another issue is that, in most works, cell phenotyping is based on immunological markers specific to neural cell types and subtypes, and thus limits the resolution of the study to the level of cell populations. Expectedly, the single-cell RNA analysis approach will become more and more widespread and will make it possible to determine the homogeneity or heterogeneity of these populations and to evaluate whether the culture conditions favour the emergence of certain cell clusters. Lastly, the effect of the cell population of origin should not be underestimated or overlooked in the era of “personalized” iPSCs culture: the impact of a biomaterial may differ on cells derived from different donors and from control subjects as compared to patients with neurological conditions. This has to be taken into account in the design of the model and the experimental tests. This question will undoubtedly be the subject of future investigations and findings.

Furthermore, a large body of work strongly suggests that the presence of collagen, fibronectin, HA and laminin, or a combination of them, is beneficial to neural tissue engineering (Figure 2). However, while bioink is intended to provide the most favourable environment for cells at an early stage, the cells are expected to sustainably produce and secrete in turn their own ECM and ECM regulators, such as growth factors and specific proteases, into the organoid. For example, secretion of proteases (matriptase and matrix metalloproteases 2 and 9), fibronectin, laminin and collagen was observed in 3D cultures of neural stem cells derived from ESCs, indicating their ability to remodel their extracellular environment [104]. The cellular ontology is too complex to consider a universal bio-ink allowing for the simultaneous differentiation of all cell lines of the central nervous system. This is illustrated by the ambivalent role of fibrin on gliogenesis and neuronogenesis. Comparable but divergent procedures to generate organoids recapitulating brain cortex, hippocampus, thalamus, hypothalamus, midbrain, cerebellum or retina have been published [147]. Three-dimensional bioprinting is particularly suitable to overcome this issue since it allows for the assembly of cells previously grown or differentiated in different biogels, at a stage where they have matured enough to be able to survive, function, communicate and secrete the mediators to support homeostasis of their different kinds.

In conclusion, a many research studies have provided significant insight on the most suitable biomaterials and blends for 3D bioprinting of human brain organoids. In particular, unmodified natural factors efficiently support a variety of 3D neural models. Optimization of specialized hydrogels and increased resolution of cell phenotype analysis will further extend the spectrum of possible bioprinted disease models. Three-dimensional bioprinting of human cerebral organoids is an emerging and ground-breaking approach that will overcome the limitations of self-organized organoids, provide new biological and pathophysiological insight, and pave the way to innovative screening approaches.

## Figures and Tables

**Figure 1 biomolecules-13-00025-f001:**
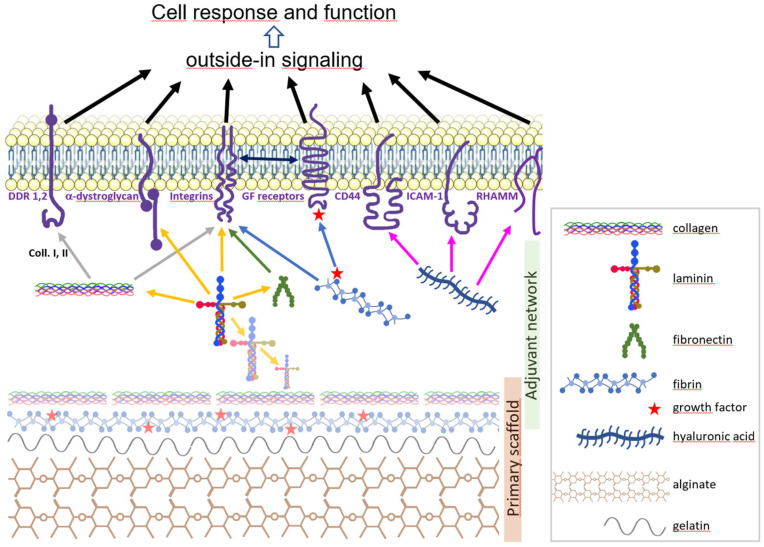
Graphic summary of a hypothetical scaffold for neural organoid bioprinting. A primary scaffold (bottom) is composed of polymers able to form a gel intrinsically (alginate, collagen, fibrin) and to enhance bioprinting performance (gelatine). A secondary network of biomolecules (middle) further structures the hydrogel and provides the cell (top) with a molecular neighbourhood close to its natural microenvironment. Shown are collagen, laminin, fibronectin, fibrin, and hyaluronic acid (left to right) which bind (color-coded arrows) to membrane receptors DDR 1 and 2, α-dystroglycan, integrins, CD44, ICAM-1 and RHAMM, which are connected to cytoskeleton and signalling factors and in turn transduce the outside-in signal to the cell. Note that integrins can bind some GF receptors. Growth factors (GF, red stars) bind to fibrin transferred to GF receptors. Laminin chains form a supramolecular network one with another, and this network is prone to bind other ECM molecules such as collagen and fibronectin.

**Figure 2 biomolecules-13-00025-f002:**
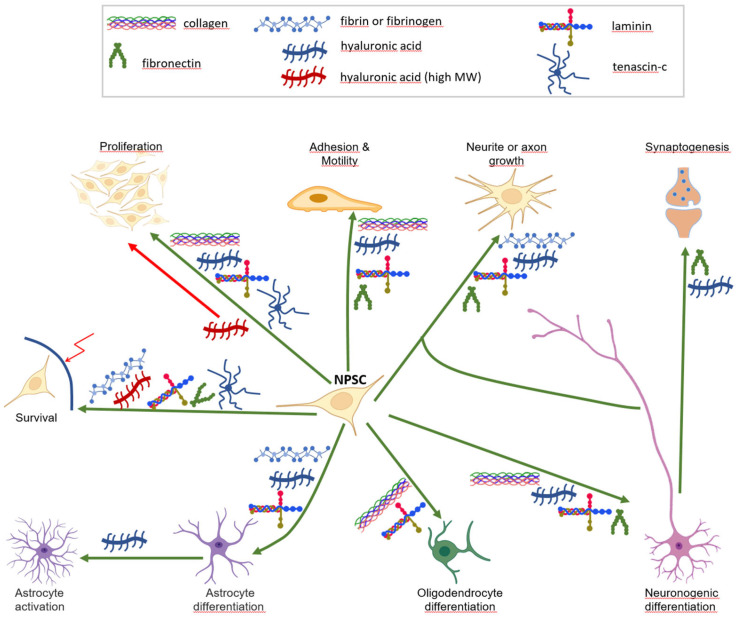
Outcomes of natural biomaterials on neural cell progenitors. Shown are the positive (green arrows) or negative (red arrows) effects of various biomacromolecules on neural stem/progenitor cell (NPSC, centre) homeostasis and functions. Note the presence of laminin in almost all branches, and the specific effect of high molecular weight HA. Some cell icons are from the Biorender gate (biorender.com).

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
