# Peer review of "Message in a Scaffold: Natural Biomaterials for Three-Dimensional (3D) Bioprinting of Human Brain Organoids"

_biomolecules, 2022, doi:10.3390/biom13010025_

Round 1

Reviewer 1 Report

The review is devoted to an important topic for any cell biologist. The authors collected information about the scaffolds and structured it competently. I believe that the article deserves publication after correcting typos in the text and adjusting the language.

Minor:

Line 117: "3.3. D" instead of "3. 3D".

Line 208: Error in abbreviation NGF decoding.

Line 216: Should be nigra instead of negra.

Reviewer 2 Report

The manuscript reviewed existing biomaterials used for constructing brain organoids.  Specially, the review focused on bioink base materials such as alginate, helper agent such as gelatin and adjuvant materials (collagen, fibrin, hyaluronic acid and laminin). The review would fill a gap in biomaterials for neural tissue engineering and bioprinting, thus worth publishing.

One major concern is the overlook of collagen’s role in constructing brain organoids.  The statement of “Since then, collagen has been almost exclusively investigated as a hydrogel base polymer in the field of peripheric nervous system, and does not seem to raise interest for 3D culture of brain cells” (page 7, line 293-295) is not true. 

Collagen has been investigated for 3D brain cell cultures for a long time.  It has been successfully used for brain tissue constructs with rodent cortical neurons (Tang-Schomer, M.D., et al., Proc Natl Acad Sci U S A, 2014. 111(38): p. 13811-6.; Chwalek, K., et al., J Vis Exp, 2015(105): p. e52970; Chwalek, K., et al., Nat Protoc, 2015. 10(9): p. 1362-73.; Sood, D., et al., ACS Biomater Sci Eng, 2016. 2(1): p. 131-140.).

Collagen is also found to support human iPSC-derived 3D neuronal cultures (William L. Cantley, et al., ACS Biomaterials Science & Engineering 2018. 4(12): p. 4278-4288.) as well as primary human brain cell cultures (Tang-Schomer, M.D., et al., J Tissue Eng Regen Med, 2018. 12(5): p. 1247-1260.).

Another aspect concerning the printablility of these biomaterials needs to be discussed.  The review is thorough in its analysis of the biological functions of the biomaterials.  However, since "bioprinting" is the focus area as indicated in both the title and abstract, a reader would expect some discussion on how these materials can be used for bioprinting.  There is an excellent paragraph on alginate for bioprinting, but discussion on other materials is missing.  It is not clear if attempts on bioprinting on these other materials (e.g., collagen, matrigel, fibrin or composite gels) have not been made or not successful or not as impressive as alginate.  Comparison and contrast with alginate in terms of printability would provide valuable insights that a reader would expect from such an important topic. 

Other minor issues:

11.    “Unvaluable” should be replaced with “invaluable” throughout the manuscript.

22.    Reference 30 is a comment on the original research article of Mansour, A., Gonçalves, J., Bloyd, C. et al. An in vivo model of functional and vascularized human brain organoids. Nat Biotechnol 36, 432–441 (2018). https://doi-org.ezproxy.lib.uconn.edu/10.1038/nbt.4127

33.    Why this sentence is in all negative sense? “In the developing organoid, the cells benefit from neither the same cell neighbourhood, nor the development factors (finely regulated in space and time), nor the mechanical constraints (as imposed for example by the developing skull) that exist in embryo and fetus” (page 3, line 100-102)

44.    “shortenage of progenitor populations” ? (page 3, line 113)

55.     “3.3. D bioprinting: benefits and challenges” ?

66.     “wrapp” should be replaced with “wrap”. (page 4, line 155-121)

77.    “caudalizing effect [46]” ? (page 4, line 169)
